# Inhibitory Effects of Chlorogenic Acid Containing Green Coffee Bean Extract on Lipopolysaccharide-Induced Inflammatory Responses and Progression of Colon Cancer Cell Line

**DOI:** 10.3390/foods12142648

**Published:** 2023-07-09

**Authors:** Atita Panyathep, Khanittha Punturee, Teera Chewonarin

**Affiliations:** 1School of Medicine, Mae Fah Luang University, Chiang Rai 57100, Thailand; atita_puy@yahoo.com; 2Cancer Research Unit of Associated Medical Sciences (AMS-CRU), Faculties of Associated Medical Sciences, Chiang Mai University, Chiang Mai 50200, Thailand; khanittha.taneyhill@cmu.ac.th; 3Department of Biochemistry, Faculty of Medicine, Chiang Mai University, Chiang Mai 50200, Thailand

**Keywords:** colorectal cancer resection, lipopolysaccharide, green coffee bean extract, chlorogenic acid, inflammatory responses, metastasis

## Abstract

An inflammatory response, related to colorectal cancer (CRC) progression, is a major subsequent result of bacterial infection following CRC surgery and should be of serious concern. Lipopolysaccharide (LPS), from the bacterial membrane, is a vital mediator of this event through binding with a Toll-like receptor 4 (TLR4) and activating through NF-κB in CRC. To identify a novel inhibitor of LPS-induced colon cancer cells (SW480), green coffee bean extract (GBE) was investigated. Ethyl acetate insoluble fraction (EIF) was mainly collected from GBE and classified as chlorogenic acid (CGA)-rich fractions. EIF and CGA inhibited TLR4 expression in LPS-induced SW480 cells. However, EIF was more dominant than CGA, via inhibition of expression and secretion of several associated mediators in inflammatory responses and CRC metastasis through NF-κB inactivation, which resulted in the abrogation of CRC migration and invasion. Thus, CGA-rich fraction from GBE can be further developed as an alternative treatment, coupled with CRC surgical treatment, to increase therapeutic efficiency and survival rate.

## 1. Introduction

In its early stages, colorectal cancer (CRC) is often cured following operative treatment. However, survival rates, following CRC resection, can be compromised, often as a result of bacterial infection during or post operation [1,2]; therefore, reducing risk factors for bacterial infection and the prevention of the serious sequelae, which results from these pathogens, should be a point of concern in CRC treatment. Lipopolysaccharide or LPS is the major contributor of several adverse events. LPS is derived from the outer part of Gram-negative bacteria, the predominant bowel pathogens. It is known that infection and inflammation are involved with tumor progression [3,4], related to subsequent LPS action, which consists of inflammatory responses leading to cancer metastasis [5,6]. We can postulate that bacterial infection, in peri- or post-CRC resection, induces inflammatory responses and is conductive to cancer recurrence and metastasis of any remaining CRC.

Normally, LPS acts on its specific receptor, named Toll-like receptor 4 or TLR4, located on the cell membrane of several cell types, especially in cancer cells. TLR4 is predominantly found in both inflamed intestinal and CRC cells [7,8], which also increases the risk of LPS trigger events. TLR4 functions as a link between bacterial infection, colonic inflammation, and cancer progression [7,9]. LPS-TLR4 enhances the production of numerous mediators associated with inflammatory responses [7,10,11], e.g., IL-6, TNF-α, IL-8, IL-1, and CRC progression, e.g., VEGFC, MMP2, MMP9, ICAM-1, VCAM-1 [10,12,13]. In particular, VEGFC has been identified as a bridge to several steps of CRC metastasis, namely: migration, invasion, lymphangiogenesis, and lymph node metastasis [9,10,14,15]. Among the many TLR4-related signaling pathways, NF-κB is predominantly involved in, not only inflammatory pathways, but also in the facilitation of CRC metastasis [12,16,17]. Therefore, the disturbance of LPS-TLR4 signaling has been considered as a therapeutic target to reduce complications from CRC resection.

In some previous reports, anti-inflammatory drugs such as aspirin and celecoxib, delayed the progression of CRC [18,19], but cardiovascular safety and other side effects remain a point of concern. Therefore, natural dietary supplements, coupled with conventional treatments, were highlighted with the intent to reduce toxicity and increase therapeutic efficiency. However, research information is quite limited and not extensive. Our recent research, relating to anti-inflammatory responses, as well as inhibition of cancer progression in LPS-induced human colon cancer SW480 cells, was clearly documented in a study using purple rice extract [20], which encouraged efforts to search for more novel and natural inhibitors.

Coffee beans have two main strains, Arabica and Robusta. Two major compounds are well-known and plentiful in coffee beans, being caffeine (CF) and chlorogenic acid (CGA), especially in green coffee beans of the Robusta strain [21]. Anti-Gram-negative pathogenic bacteria [22,23,24], anti-inflammation, and anti-CRC cancer progression of these compounds has been revealed [21,25], However, their action on LPS-TLR4 of CRC has not been researched before. Thus, this is the first report on the new role of green coffee bean extract (GBE) in the reduction of LPS-induced inflammatory responses and cancer progression of CRC cells.

## 2. Materials and Methods

### 2.1. Plant Extraction

Dried green coffee beans (*Coffea canephora* or *Coffea robusta*) were purchased from Doi Pha Hee, Chiang Rai province. They were crushed to a fine powder. Fifty grams of powder were shaken with 1 L of 60% methanol (in a ratio of 1:20 (*w*/*v*)) at room temperature. The filtrate was collected by Whatman No.1 and then concentrated to a final volume of 100 mL by the evaporator, called crude methanol extract (CME). Thereafter, CME (100 mL) was further separated with hexane to remove lipophilic fraction (in ratio 1:1) into 2 parts; hexane soluble fraction (HSF) and hexane insoluble fraction (HIF). HIF was further separated with ethyl acetate to decaffeinate (in ratio 1:1) into 2 parts; ethyl acetate soluble fraction (ESF) and ethyl acetate insoluble fraction (EIF). Following this, all of the fractions (HSF, ESF, and EIF) were evaporated and lyophilized. Each fraction had its yield calculated as % yield and further chemical composition analysis was performed.

### 2.2. Chemical Analysis

#### 2.2.1. Total Phenolic Acid Content

Total phenolic content was examined in accordance with Folin–Ciocalteu method [26]. In short, various concentrations of standard gallic acid (0–0.32 mg/mL) and all fractions at 1 mg/mL were mixed with Folin–Ciocalteu reagent, distilled water and 7% sodium carbonate (Na_2_CO_3_), respectively. Each mixture was incubated at 45 °C for 15 min and then its absorbance was measured at 765 nm using a spectrophotometer. A standard curve of gallic acid was created and used to calculate the phenolic content of each fraction at 1 mg/mL. The results are shown as mg of the gallic acid equivalent (GAE) per 1 g of the dry weight sample.

#### 2.2.2. Total Flavonoid Content

A colorimetric assay with aluminum chloride was utilized for quantifying total flavonoid content [26]. In brief, several concentrations of standard flavonoid or catechin (0–0.32 mg/mL), with all fractions at 1 mg/mL, were allowed to react with 5% sodium nitrite (NaNO_2_), distilled water, 10% aluminium chloride (AlCl_3_·6H_2_O) and 1M sodium hydroxide (NaOH), respectively. After 5 min, each sample was measured with a spectrophotometer; absorbance at 532 nm. Finally, a standard curve of catechin was formed and total flavonoid content of each fraction at 1 mg/mL was computed. These results are recorded as mg of catechin equivalents (CAE) per 1 g of the dry weight sample.

#### 2.2.3. High Performance Liquid Chromatography

Two different methods were used for quantifying CGA, as well as CF, along with other phenolic acids, in all the fractions from GBE. For CGA detection, each fraction was separated on column ultra C18, 5 µM, 250 × 4.6 mm (Restex, Schertz, TX, USA) using 0.2% phosphoric acid in water and acetonitrile as the mobile phase. A UV detector (DAD) was set at 330 nm. For CF and other phenolic acids detection, column platinum EPS C18 100A, 3 µM, 53 × 7 mm (Grace, Columbia, MD, USA) and mobile phase 0.05% trifluoroacetic acid in 13% acetonitrile were used with the UV detector at 210 nm. Both methods were set at a flow rate of 1 mL/min. The results were displayed as HPLC chromatograms and identified by comparison with retention times (RT) of standard CGA, CF, and other phenolic acids (gallic acid, catechin gallate, epicatechin, gallocatechin, and epicatechin gallate). In the final analysis, the content of CGA, CF, and other phenolic acids were computed by using the standard curve of each standard (Sigma Aldrich, St. Louis, MO, USA) and shown as mg/100 g extract.

### 2.3. Antioxidant Assays

#### 2.3.1. DPPH Scavenging Assay

The scavenging ability of each fraction to 2, 2-diphenyl-1-picrylhydrazyl or DPPH (Sigma Aldrich, St. Louis, MO, USA) was initiated with varying concentrations of each fraction (25–100 µg/mL). Each sample was incubated with 2 mM DPPH at room temperature for 30 min in darkness. The absorbance was detected at 517 nm and computed as the percentage of scavenging of DPPH by comparison with a blank control (100% DPPH or 0% scavenging ability). The blank was methanol instead of a sample. The results were expressed as a concentration of the sample at 50% scavenging of DPPH (SC_50_).

#### 2.3.2. Iron Chelation Assay

The chelating of iron was also referred to anti-oxidant activity. Briefly, the different concentrations (100–500 µg/mL) of each fraction were set. Each concentration was incubated with 2 mM ammonium iron (II) sulfate solution and 5 mM ferene. After 10 min at room temperature, the color reaction was measured at 593 nm and calculated as the percentage of iron chelation (vs. blank control). The blank was methanol instead of a sample (100% iron or 0% iron chelation). A concentration of the sample at 50% iron chelation (EC_50_) was reported.

### 2.4. Cell Culture

Human colon adenocarcinoma cell lines or SW480 cells (ATCC, Rockville, MD, USA) were chosen for the in vitro model. They were cultured in 10% fetal bovine serum (FBS) contained in Dulbecco’s Modified Eagle Medium (DMEM) (Invitrogen Corporation, Bohemia, NY, USA) under proper conditions (37 °C and 5% CO_2_ within an incubator). Normal human dermal fibroblast cells (NHDF) (ATCC, Rockville, MD, USA) were used as the normal cell control and their growth condition was identical to the SW480 cells. SW480 cells were thoroughly pretreated with 1 µg/mL of LPS (in serum-free media) derived from *Escherichia coli* O26:B6 (Sigma Aldrich, St. Louis, MO, USA) for 4 h.

### 2.5. Cell Viability Assay

SW480 cells (at 5 × 10^3^ cells/well in 96-well plate) were incubated at 37 °C for 24 h before being pretreated with LPS (1µg/mL for 4 h). The supernatant was removed and substituted with various treated conditions. After 24 h at 37 °C, cell viability or toxicity was detected by 3-(4,5-Dimethylthiazol-2-yl)-2,5-diphenyltetrazolium bromide or MTT assay (Sigma-Aldrich, St. Louis, MO, USA). MTT solution (5 mg/mL) also supplanted the prior condition media and was further placed in 37 °C for 2 h. The color was developed by adding dimethyl sulfoxide (DMSO). The absorbance was quantified at 540 nm, which allowed computation as the percentage of cell viability (vs. LPS control). Correspondingly, for verifying the toxicity to normal cells (NHDF), these were kept under identical conditions as the SW480 cells (under without LPS condition).

### 2.6. Quantitation of Secretory Proteins

The secretion of associated proteins was measured by this specific sandwich ELISA kit (Sigma Aldrich, St. Louis, MO, USA). The sample was conditioned media from each treated (EIF or CGA) condition on LPS-pretreated SW480 cells for 24 h. All samples were processed by following the package insert, and the absorbance was measured at 450 nm and computed by comparison with a specific standard curve (pg/mL), as well as the total protein content (mg). Each secretory protein content (pg/mL per mg of total protein content) was finally shown as the percentage of the secretory protein content (vs. LPS control)

### 2.7. Immunodetection of Intracellular Proteins

An immunoblotting assay was used for analyzing the expression level of several associated mediators of LPS-TLR4 signaling; these included TLR4, TNF-α, IL-1β, COX-2, VEGFC, pp65, and p65. In short, the cell lysate, from various treated conditions on LPS-pretreated SW480 cells (at 37 °C for 24 h), was equally loaded onto 10% sodium dodecyl sulfate-polyacrylamide gel electrophoresis (SDS-PAGE). Thereafter, gels were blotted onto polyvinylidene difluoride (PVDF) membranes, which were continuously blocked by 5% skimmed milk powder in PBS-T (0.05% Tween-20 in PBS). Afterward, membranes were incubated with diluted specific primary antibodies against TLR4, TNF-α, IL-1β, COX-2, pp65, p65, VEGFC, and β-actin (Santa Cruz, Rio Grande, TX, USA and Merck Millipore, Burlington, MA, USA) at 4 °C overnight. For detection, membranes were soaked with diluted HRP-conjugated secondary antibody (Merck Millipore, Burlington, MA, USA) at room temperature and further reacted with the enhanced chemiluminescence (ECL) substrate (Bio-Rad^®^ Laboratories, Hercules, CA, USA). The bands were visualized and captured using a chemiluminescence detector (Bio-Rad^®^ Laboratories, Hercules, CA, USA). Finally, the intensity of each band was analyzed and normalized with its β-actin band.

### 2.8. Cell Adhesion Assay

LPS-pretreated SW480 cells (at 50 × 10^4^ cells/mL in 1% BSA in serum-free medium) were suspended with each concentration of EIF or CGA in a ratio of 1:1 at room temperature. Then, each mixture (100 µL) was added into 50 µg/mL of Matrigel (Corning^®^, Corning, NY, USA) coated 96-well plates (in triplicate), and then placed at room temperature for 1 h. Non-adherent cells were removed and replaced with MTT solution for measuring the remaining cell-Matrigel adhesion. The absorbance was detected and calculated as a percentage of cell adhesion ability.

### 2.9. Cell Migration Assay

#### 2.9.1. Wound Healing Assay

This method is commonly used to verify the effect of compounds on cell migration by measuring the distance of cell travel (2D movement). To start, LPS-pretreated SW480 cells (at 2 × 10^6^ cells/well) were seeded into 6-well plates at 37 °C for 24 h and then the media was removed. Each conditioned media was added and then the gap (at 0 h) was created by the pipette tip. The migratory distance was measured after incubation at 37 °C for 24 h, by using an inverted microscope (10×) and Zen imaging software. The picture of each condition (at 0 and 24 h) and its percentage of cell migration (vs. LPS control) was computed and presented.

#### 2.9.2. Transwell Migration Assay

This migration is dependent on the concentration gradient of the chemoattractant (3D movement). Firstly, 100 µL of the mixture of LPS-pretreated SW480 cells at 2 × 10^6^ cells/mL in serum-free media (0% FBS in DMEM), and each concentration of EIF or CGA (in ratio 1:1), was placed on the upper part of the Transwell^®^ with an 8 µM-pore-polycarbonate membrane (in duplicate). The lower part of the chamber contained 600 µL of 10% FBS in DMEM. The cell motion at the lower part of the membrane was detected after incubation at 37 °C for 24 h by staining and fixing with 1% crystal violet in 50% methanol. Non-motion cells were removed with a cotton swab. Each membrane was cut and dissolved with 20% acetic acid. The absorbance was measured at 570 nm, which directly correlated with cell migratory ability. The data was expressed as a percentage of cell migration by comparison with LPS control (100% cell migration).

### 2.10. Cell Invasion Assay

The cell invasion assay, or Transwell migration assay, was performed by coating Matrigel (basement membrane matrix) on the membrane. Similarly, the mixture of LPS-pretreated SW480 cells (at 2 × 10^6^ cells/mL) and each treated condition (in a ratio of 1:1) in serum-free media was placed into the upper chamber (in duplicate). In the lower part, 10% FBS in DMEM was also added. After 24 h, the cell invasion was detected by staining and fixing with 1% crystal violet in 50% methanol. Non-invasive cells were cleared with a cotton swab. The absorbance of each membrane in 20% acetic acid was measured at 570 nm. The percentage of cell invasion (vs. LPS control) was displayed.

### 2.11. Statistical Analysis

All experiments were independently repeated at least three times. The results were shown as means ± SD and the significant values were analyzed by using one-way analysis of variance (ANOVA) via the SPSS program. The major symbols were determined as significant values (vs. LPS control), including ^a^ *p* ≤ 0.05 and ^b^ *p* ≤ 0.005.

## 3. Results

### 3.1. Chemical Contents and Antioxidant Activities

Four main fractions were collected from green coffee bean extract, consisting of CME, HSF, EIF and ESF. Table 1 and Table 2 show their percentage of yield (vs. 50 g dry weight), total phenolic (TPC) and flavonoid content (TFC). As expected, the results of % yield from TPC and TFC were the highest in CME (primary fraction of GBE), followed by its separated fractions; EIF, ESF and HSF, respectively. Indeed, CGA and CF were the main ingredient of GBE, so these compounds were also found in all fractions by using HPLC and comparison with standard CGA (Figure 1A) and CF (Figure 1F). CGA content was plentiful in CME and EIF (17,768.36 and 17,683.78 mg/100 g, respectively), but hardly seen in ESF and HSF (Figure 1B–E). On the other hand, ESF was richest in CF content (61,043 mg/100 g) and more than CME and EIF, approximately 8–9 times higher (Figure 1G–J). However, only a small amount of other phenolic acid compounds, such as catechins and gallic acid, were found in all fractions (Table 3). Thus, ESF and EIF were interesting due to the high concentrations of the major compounds of GBE, also called CF-rich fraction and CGA-rich fraction, respectively. Moreover, both had their antioxidant activities confirmed using different methods. The scavenging capacity of ESF and EIF were different, which seemed to rely on chemical ingredients and the method used. ESF was stronger in DPPH scavenging (less SC_50_), while EIF was more efficient in iron chelation (less EC_50_) (Table 2).

### 3.2. Effect of EIF, ESF, CGA, and CF on Colon Cancer Cell Viability under w/wo LPS Stimulation

Our results showed no toxicity in the used dosage ranges of all tested substances to SW480 cell survival, in both the with and without LPS-pretreated conditions (Figure 2A–F). Interestingly, SW480 cell proliferation was gradually reduced with ESF treatment (25–200 µg/mL) under no LPS condition (Figure 2A), but this was not found in LPS-pretreated SW480 cells (Figure 2D). Finally, these results also revealed the optimal concentration (inhibitory concentration at less than 20% of cell viability or <IC_20_) range of each substance, including 25–75 µg/mL of EIF and ESF, 5–20 µg/mL of CGA and 20–60 µg/mL of CF. Of note, these concentrations of each substance were also shown to be nontoxic to the cell viability of normal fibroblast cells or NHDF (normal cell control) (Figure 2G–I).

### 3.3. Effect of EIF, ESF, CGA, and CF on TLR4 Expression of LPS—Induced Colon Cancer Cells

TLR4 is the primary target of LPS stimulation. LPS acts as an inducer of TLR4 expression on SW480 cells (approximately 15–20% of induction in this study). We found interesting results regarding substances that could affect this LPS-TLR4 pathway. Not only did EIF treatment dominantly downregulate TLR4 expression on LPS-induced SW480 cells in a dose-dependent pattern, especially at 50 and 75 µg/mL (Figure 3A,B), but also CGA treatment, having similar results. In contrast, LPS-induced TLR4 expression was constant in ESF and CF treatment (Figure 3C,D).

### 3.4. Effect of EIF and CGA on the Expression and Secretion of Inflammatory Mediators of LPS-Induced Colon Cancer Cells

Increased levels of inflammatory mediator expression and secretion are the result of the action of the LPS-TLR4 trigger. TNF-α, IL-1β, IL-6, and COX-2 expression and secretion were increased in LPS-activated cells. These LPS actions were clearly disrupted, not only expression (COX-2, TNF-α, and IL-1β) (Figure 4A–D) but also secretion (TNF-α, IL-1β, and IL-6) (Figure 4E–G) by EIF- and CGA-treated conditions, especially EIF at 25–75 µg/mL. Noticeably, all the used concentrations of CGA significantly reduced IL-1β expression but the inhibitory tendency was only gradually decreased (down to 18% from 5 to 20 µg/mL) (Figure 4D). Summarily, broad and strong anti-inflammatory responses of EIF were present, while CGA seemed to be a part of these diminished effects of EIF (but not all).

### 3.5. Effect of EIF and CGA on the Expression and Secretion of VEGFC of LPS-Induced Colon Cancer Cells

VEGFC is one of the main secretory proteins associated with various steps of colon cancer progression. Here, the cancer-inducing property of LPS has been demonstrated by the increase of VEGFC expression (18%) and secretion (40%) from SW480 cells, whereas its inductions were inhibited after adding EIF or CGA in a dose-dependent action (Figure 5A–C).

### 3.6. Effect of EIF and CGA on Adhesion, Migration, and Invasion Ability of LPS-Induced Colon Cancer Cells

Many colon cancer progression steps were examined, namely: adhesion, migration, and invasion. LPS at 1 µg/mL was enough to trigger adhesion, migration, and invasion. Surprisingly, LPS-induced SW480 cell adhesion was unchanging in both EIF and CGA testing environments (Figure 6A). On the contrary, LPS-induced SW480 cell migration under *w*/*wo* chemoattractant condition was significantly abrogated by EIF treatments, especially at 75 µg/mL. Meanwhile, anti-migration of CGA was selectively found, without a chemoattractant condition (2D movement) in a dose-dependent pattern (Figure 6B–D). Correspondingly, LPS-stimulated SW480 cell invasion under the chemoattractant gradient was specifically diminished by treatment with EIF (but not CGA), especially EIF at 75 µg/mL (34%) (Figure 6E).

### 3.7. Effect of EIF and CGA on NF-κB Activation of LPS-Induced Colon Cancer Cells

In NF-κB activation, LPS seemed to prefer phosphorylation of p65 (around 20% of activation) to upregulation of p65 levels in SW480 cells, compared with no LPS condition; this is similar to our previous study [20]. Interestingly, EIF treatment obviously diminished both pp65 (38%) and p65 (34%) levels, even when there was no LPS response to p65, especially at EIF levels of 50 and 75 µg/mL (Figure 7A–C). On the contrary, there were no distinct effects on pp65 and p65 levels after CGA treatment. Correspondingly, the fold change of pp65/p65 of the LPS control seemed to slightly increase (vs. no LPS condition, *p* > 0.05), while the other treated conditions showed no significant differences in the ratio of pp65/p65 (vs. LPS control), which is definitely related to their actions on pp65 and p65 levels (Figure 7D).

## 4. Discussion and Conclusions

Even though the adverse effects of infection with pathogenic bacteria during CRC surgery are well known, there is still limited investigation, other than antibiotic therapy, related to other agents to minimize these serious events. Our previous publication was first able to identify the inhibitory role of a gamma oryzanol-rich fraction from purple rice on LPS-induced inflammatory responses and CRC progression, via pathways of TLR4 and NF-κB [20,27,28]. Here, our second achievement was to identify similar properties in green coffee bean extract or GBE, in order to assess increased survival rates of CRC patients after surgery. In this study, GBE was finally separated into two main fractions using ethyl acetate: these being ESF (CF-rich fraction) and EIF (CGA-rich fraction). Ethyl acetate is the proper solvent to decaffeinate green coffee beans and has been approved by US FDA since 1982 [29,30]. Additionally, both fractions also contained small amounts of other phenolic acid compounds, such as gallic acid, catechin gallate, and gallocatechin gallate. However, these chemical compounds were hardly found in the HSF fraction. Antioxidant activity was verified by different methods, DPPH scavenging and the iron chelating assay. ESF was dominant in DPPH scavenging, whereas iron chelation was stronger in EIF. This means that different chemical compounds might participate in the capacity of free radical scavenging. In the in vitro study, pretreated CRC with LPS at 1 µg/mL for 40 min to 24 h was enough to stimulate inflammatory responses and several metastatic steps, but not CRC proliferation, which also correlated with the LPS-treated condition in the present study (1 µg/mL for 4 h) [10,12,20,27,31,32]. It seems that LPS at low concentrations and short time frames is enough to express its virulent action. Notably, our results showed no toxicity of EIF, ESF, CGA, and CF on CRC survival, either with or without LPS presence, over a range of concentrations. Of note, though ESF was the only substance affecting CRC proliferation without LPS treatment, its toxicity did not exceed 20% (vs. control). The optimal dose ranges of EIF and ESF-treated SW480 cells were 25–75 µg/mL, which also displayed non-toxicity to NHDF (normal cell control).

Because TLR4 is the primary target of LPS and highly expressed in CRC, our results explored that LPS-induced TLR4 expression in SW480 cells was obviously suppressed by EIF and CGA treatment, but not ESF and CF. Thus, EIF and CGA were more interesting than ESF and CF; both need further studying regarding LPS-TLR4 sequelae. In this step, the inhibitory effect of EIF seemed to be stronger than CGA, because the effective dose of CGA was only performed at 20 µg/mL with less effect than EIF at 75 µg/mL (containing CGA 13.3 µg/mL).

Next, inflammatory responses are common subsequences of LPS-TLR4; the increased expression and secretion of several inflammatory mediators are found [10,20,31]. Similarly, our present results showed that LPS induced the expression of COX-2, TNF-α, and IL-1β, as well as secretion of TNF-α, IL-1β, and IL-6. These LPS consequences were completely diminished by EIF and CGA. Noticeably, the capacity of CGA on anti-LPS-induced IL-1β expression was gradually decreased via increasing the doses, so in this case, should be regarded in the usage of a pure compound.

During CRC progression, LPS was reported as a potent inducer of CRC metastasis via VEGFC modulation [12], so the suppression of VEGFC production is the main therapeutic target for CRC [14,33,34]. LPS-induced VEGFC expression and secretion were also detailed in this study, the same as a previous publication [12], which confirmed suppression by EIF and CGA; in this, CGA seemed to participate in EIF inhibitory action. Continually, we investigated the CRC metastatic process related to LPS responsiveness, namely cell adhesion, migration, and invasion. In this study, LPS played a key role as a potent enhancer of CRC adhesion, migration, and invasion. Among these LPS actions, anti-LPS stimulated cell adhesion was absent in both EIF and CGA, which were irrelevant to their anti-LPS-induced VEGFC production. As expected, there is no relative evidence of VEGFC and adhesive ability of CRC, which is also related with the action of EIF and CGA in this study. Conversely, anti-LPS stimulated migration and invasion of EIF were closely related to its action on VEGFC production. Interestingly, CGA was hardly expressed in the role of anti-LPS-induced CRC metastasis, except migration (2D movement) and VEGFC. These results confirm that anti-LPS-induced CRC metastasis of EIF involves VEGFC. Previous data have shown the interplay of VEGFC and COX-2 to cancer metastasis and inflammation, which could be interrupted by using NSAIDs [35,36,37,38]. Although the effects of NSAIDs have been reported in several clinical trials, the side effects or drug resistance of NSAIDs consumption should be considered. Thus, EIF was more interesting because of its broad inhibitory actions by disrupting several mediators in LPS-TLR4, as well as VEGFC and COX-2.

Finally, to clarify an associated regulatory pathway, NF-κB is the major downstream regulator of the LPS-TLR4 signaling pathway and also connects with VEGFC expression. Our previous study found that LPS, preferably activated via phosphorylation of p65, but not p65 levels in CRC [20,31], correlates well with this second study. In EIF action, both pp65 and p65 levels were strongly negated, while the fold change of pp65/p65 (vs. LPS control) was constant, which implies that p65 was the main target of EIF and resulted in the reduction of pp65. In the case of CGA, both pp65 and p65 levels remained unchanged, as well as the fold change of pp65/p65, which suggests that its action could be regulated through other pathways besides NF-κB, such as JNK [12]; therefore, further study should be considered.

In conclusion, this study encourages the potential utilization of EIF or CGA-rich fraction from GBE as part of a novel, additive CRC treatment. Although, CGA is a major component and participated in some inhibitory events, it was not dominant. The synergistic effect of the mixture of compounds in EIF was highlighted, as were their impact on LPS responsiveness of CRC, namely inflammatory response, and CRC progression. Thus, it should be considered and further developed to be a safe and efficient adjunct coupled with current CRC resection in the future.

## Figures and Tables

**Figure 1 foods-12-02648-f001:**
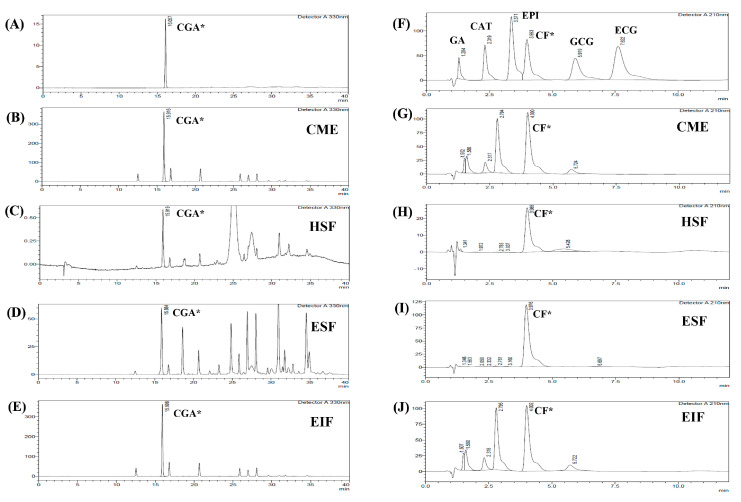
HPLC chromatogram of each fraction from GBE for identifying CGA and CF. Four fractions from GBE; CME (**B**,**G**), HSF (**C**,**H**), ESF (**D**,**I**), and EIF (**E**,**J**) were identified by comparison with RT of standard chlorogenic acid (**A**) and RT of standard various phenolic acid compounds and caffeine (**F**). Chlorogenic acid: CGA*, caffeine: CF*, gallic acid: GA, catechin gallate: CAT, epicatechin: EPI, gallocatechin gallate: GCG, and epicatechin gallate: ECG.

**Figure 2 foods-12-02648-f002:**
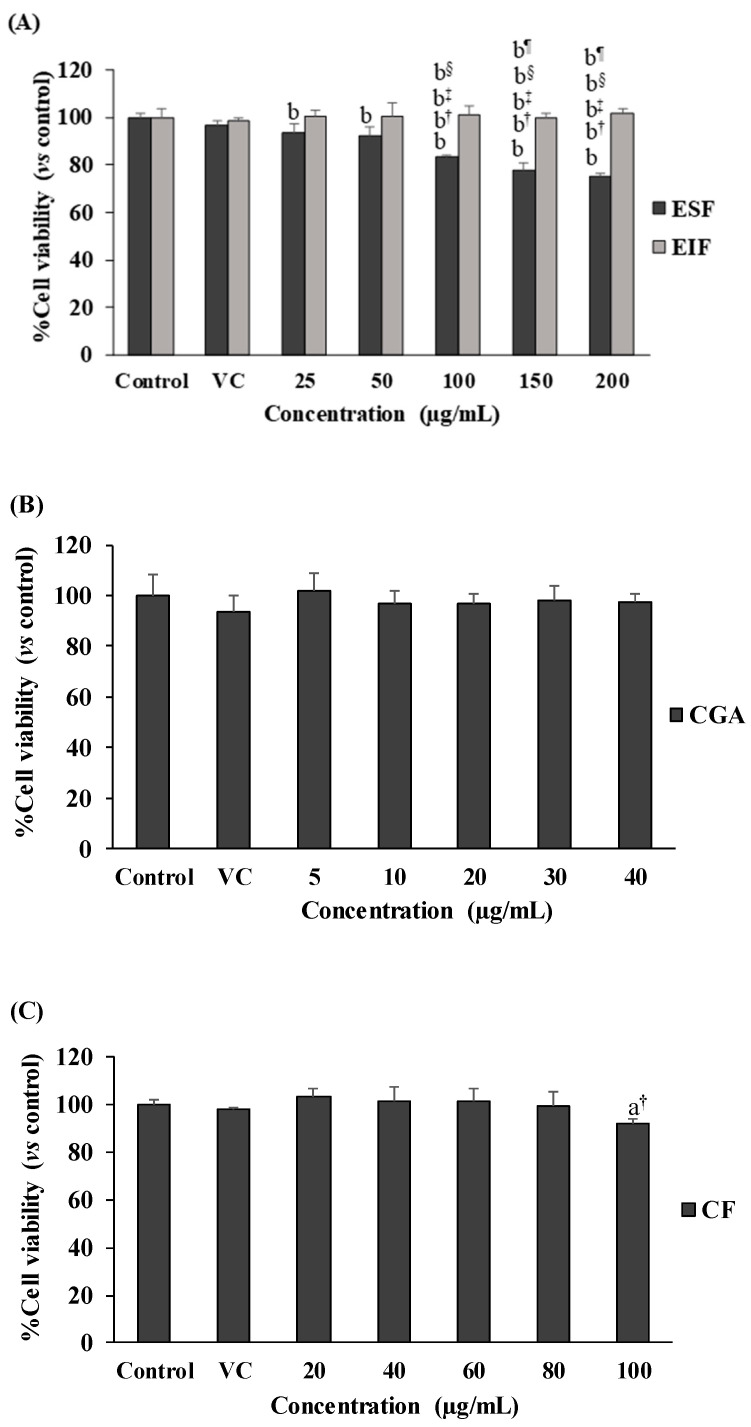
Effects of EIF, ESF, CGA, and CF on cell viability. SW480 cells were tested with various concentrations of EIF, ESF, CGA, and CF under without LPS (**A**–**C**) or with LPS (**D**–**F**) condition, then evaluated cell viability by using the MTT assay at 24 h. NHDF cells (normal cell control) were also treated with several doses of EIF, ESF, CGA, and CF without LPS condition (**G**–**I**). Letters a, *p* ≤ 0.05 and b, *p* ≤ 0.005, above the bars represent significant values of each treated condition vs. control/LPS control. Moreover, the additive symbols; †, ‡, § and ¶, on letter b, *p* ≤ 0.005, are also used to significantly compare between treated conditions; VC, 25, 50, and 100 µg/mL, respectively.

**Figure 3 foods-12-02648-f003:**
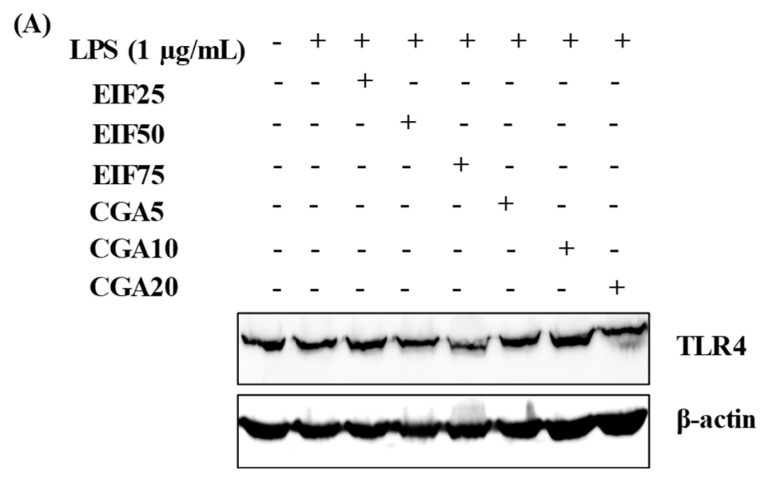
Effects of EIF, ESF, CGA, and CF on TLR4 expression of LPS-induced SW480 cells. Cell lysates of EIF (**A**,**B**), ESF (**C**,**D**), CGA (**A**,**B**), and CF (**C**,**D**) treated conditions on LPS-pretreated SW480 cells were collected and detected TLR4 levels by using Western blot. Each band intensity was quantified and displayed as mean ± SD, *n* = 3. Letters a, *p* ≤ 0.05 and b, *p* ≤ 0.005 above the bars represent significant values of each treated condition vs. LPS controls. Meanwhile, the additive symbol (†) on letters (a, *p* ≤ 0.05 and b, *p* ≤ 0.005) is also used to significantly compare between treated conditions; 25 µg/mL of EIF.

**Figure 4 foods-12-02648-f004:**
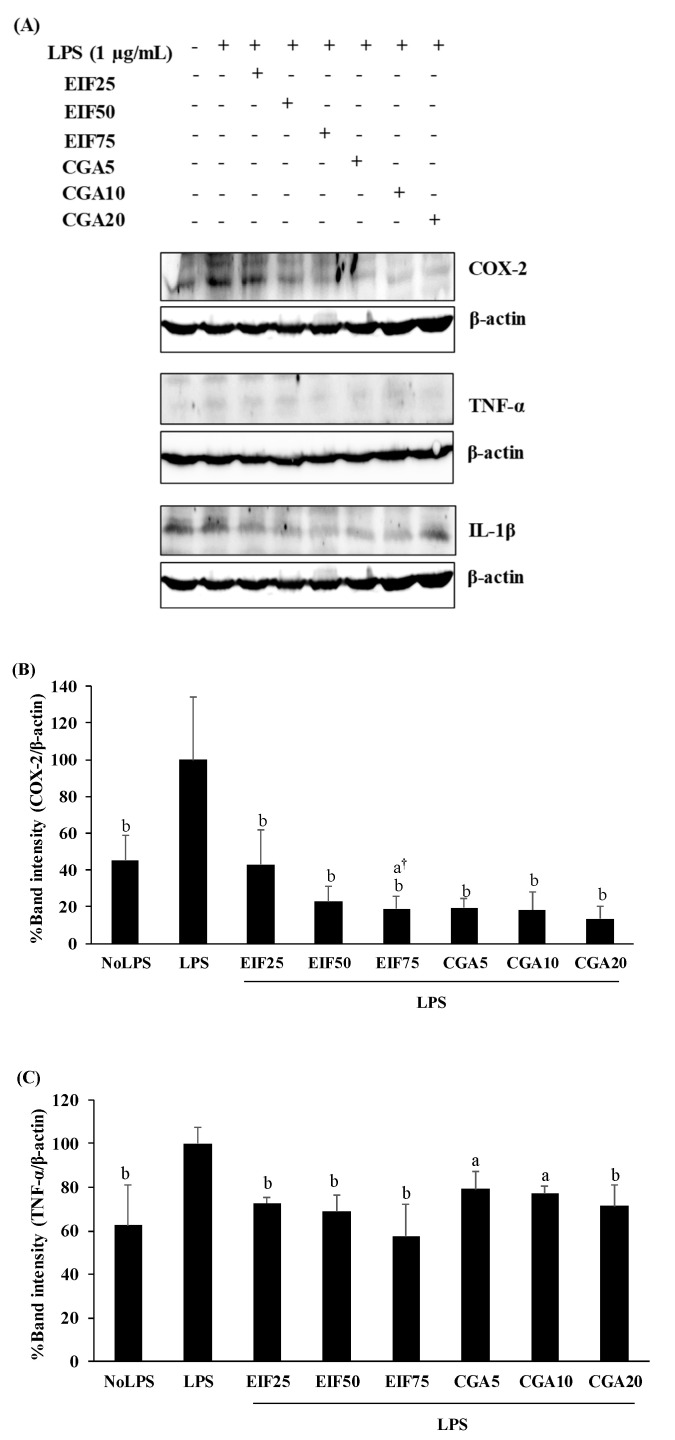
Effects of EIF and CGA on the expression and secretion of inflammatory mediators of LPS-stimulated SW480 cells. Cell lysates of EIF- and CGA-treated conditions on LPS-pretreated SW480 cells were collected, and COX-2 (**A**,**B**), TNF-α (**A**,**C**), and IL-1β (**A**,**D**) expression was measured by using Western blot. The band intensity was analyzed. Meanwhile, their conditioned media were also collected and quantified TNF-α (**E**), IL-1β (**F**), and IL-6 (**G**) secretion by using ELISA. All data were shown as mean ± SD, *n* = 3. Letters a, *p* ≤ 0.05 and b, *p* ≤ 0.005, above the bars represent significant values of each treated condition vs. LPS controls. In addition, the additive symbols; † and §, on letter a, *p* ≤ 0.05, are also used to significantly compare between treated conditions; 25 µg/mL of EIF and 5 µg/mL of CGA, respectively.

**Figure 5 foods-12-02648-f005:**
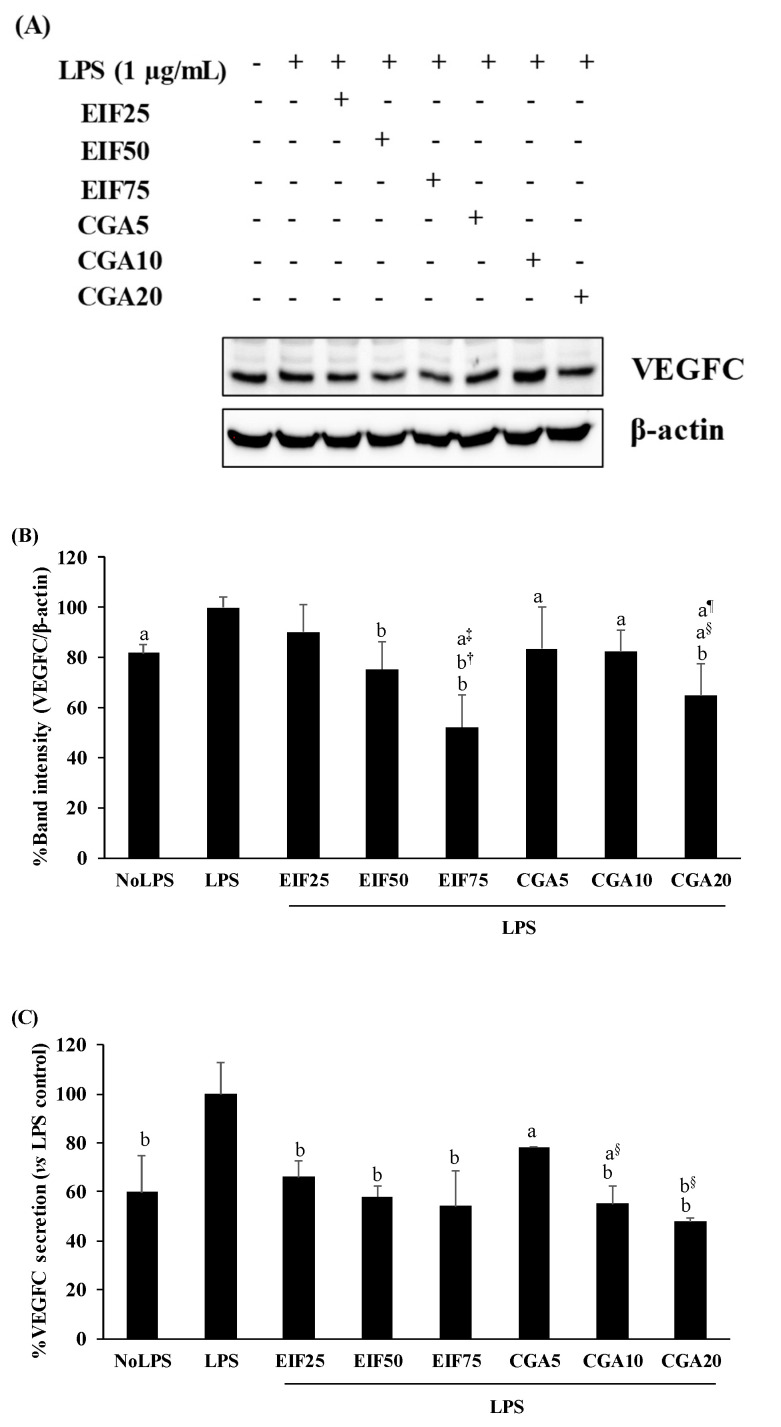
Effects of EIF and CGA on the expression and secretion of VEGFC of LPS-stimulated SW480 cells. Cell lysates and conditioned media of EIF and CGA treated conditions on LPS-pretreated SW480 cells were collected and used to examine the expression (**A**,**B**) and release (**C**) of VEGFC by using Western blot and the ELISA assay, respectively. All data were shown as mean ± SD, *n* = 3. Letters a, *p* ≤ 0.05 and b, *p* ≤ 0.005 above the bars represent significant values of each treated condition vs. LPS controls. In addition, the additive symbols on these letters (a, *p* ≤ 0.05 and b, *p* ≤ 0.005) are also used to significantly compare between treated conditions; (†) vs. 25 µg/mL of EIF, (‡) vs. 50 µg/mL of EIF, (§) vs. 5 µg/mL of CGA, and (¶) vs. 10 µg/mL of CGA.

**Figure 6 foods-12-02648-f006:**
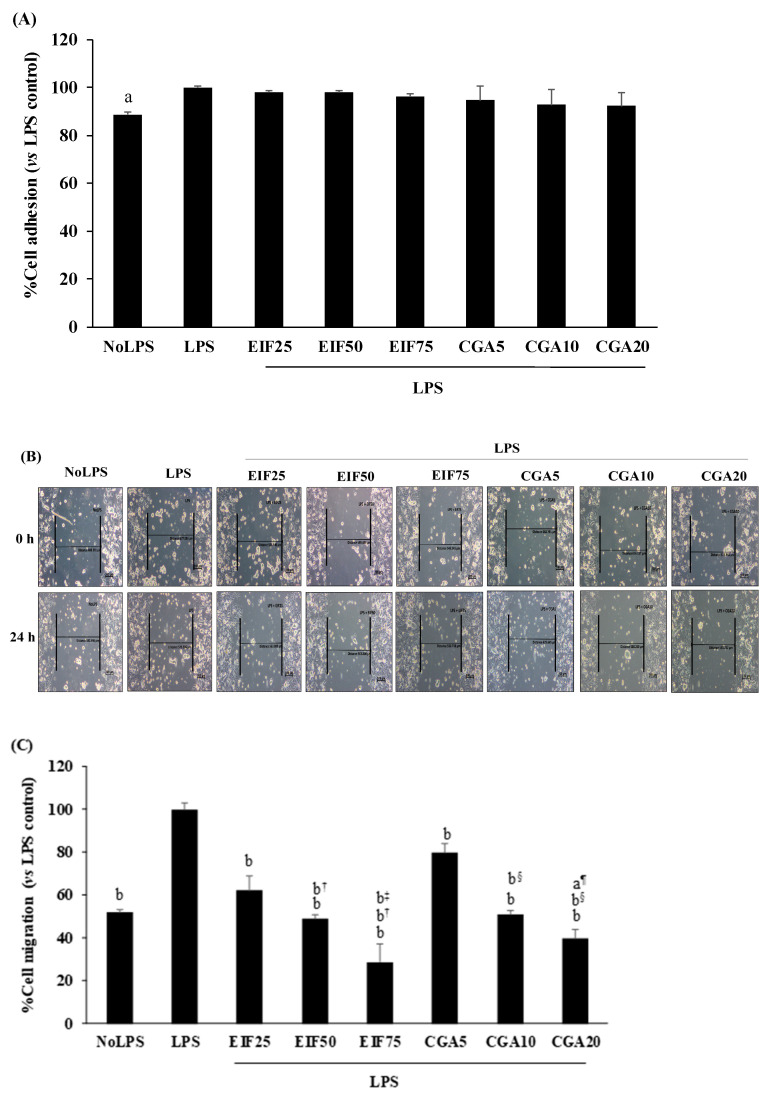
Effects of EIF and CGA on adhesion, migration, and invasion of LPS-induced SW480 cells. LPS-pretreated SW480 cells were treated with EIF and CGA at various concentrations to evaluate CRC metastatic steps; adhesion, migration, and invasion. Their adhesive ability was shown by adhesive assay (**A**). Their migratory activity was examined by wound healing (**B**,**C**) and Transwell migration assay (**D**), while invasive ability was also analyzed by Transwell invasion assay (**E**). All data were shown as mean ± SD, *n* = 3. Letters a, *p* ≤ 0.05 and b, *p* ≤ 0.005 above the bars represent significant values of each treated condition vs. LPS control. In addition, the additive symbols on these letters (a, *p* ≤ 0.05 and b, *p* ≤ 0.005) are also used to significantly compare between treated conditions; (†) vs. 25 µg/mL of EIF, (‡) vs. 50 µg/mL of EIF, (§) vs. 5 µg/mL of CGA, and (¶) vs. 10 µg/mL of CGA.

**Figure 7 foods-12-02648-f007:**
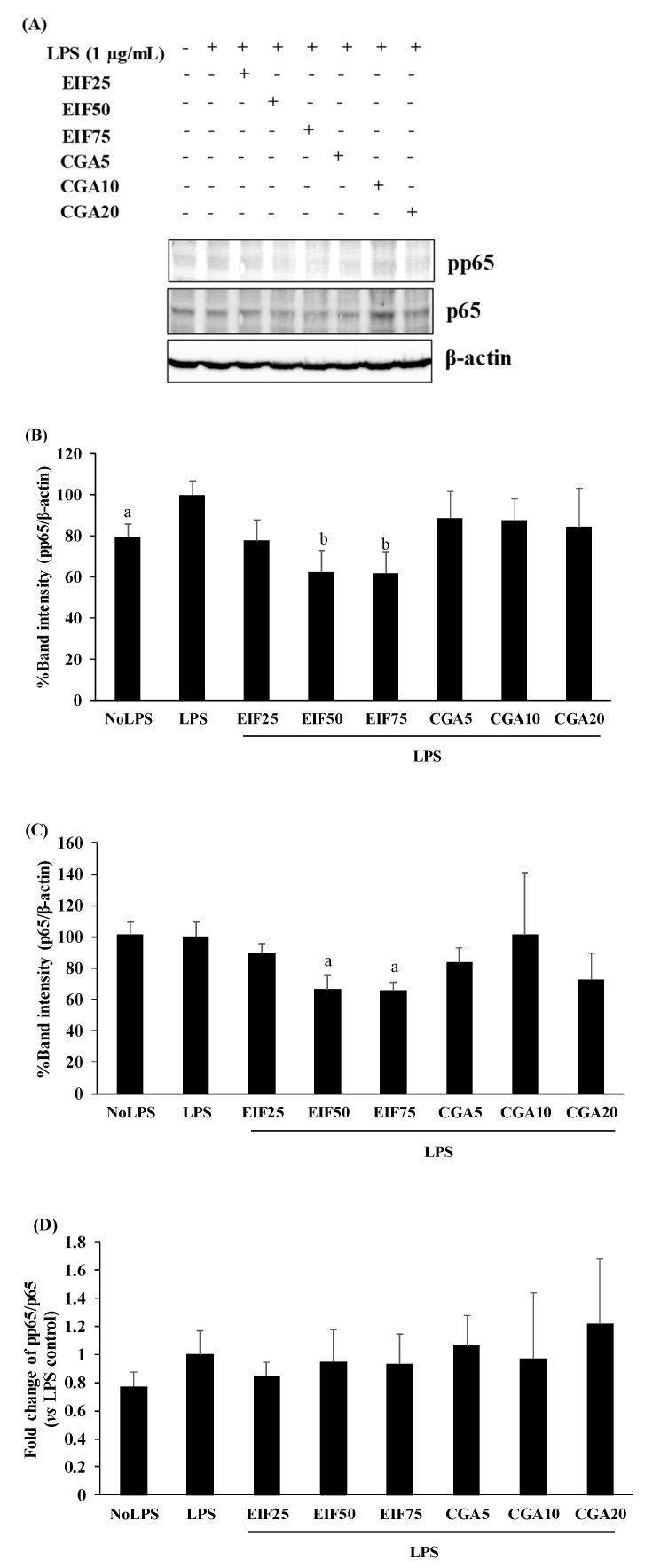
Effects of EIF and CGA on NF-κB activation of LPS-induced SW480 cells. Western blot was used for determining pp65 (**A**,**B**) and p65 (**A**,**C**) levels on LPS-induced SW480 cells after treatment with EIF and CGA at several concentrations. All band intensities were quantified and expressed as mean ± SD, *n* = 3. A fold change of pp65/p65 (vs. LPS control) was also displayed (**D**). Letters a, *p* ≤ 0.05 and b, *p* ≤ 0.005, above the bars represent significant values of each treated condition vs. LPS control.

**Table 1 foods-12-02648-t001:** Percentage of yield of CME, HSF, ESF and EIF (vs. 50 g dry weight GBE).

Green Coffee Bean Extract (GBE)	% Yield (vs. 50 g of Dry GBE)
Crude methanol extract (CME)	23.92
Hexane soluble fraction (HSF)	0.08
Ethyl acetate soluble fraction (ESF)	0.18
Ethyl acetate insoluble fraction (EIF)	17.74

**Table 2 foods-12-02648-t002:** Total phenolic content (TPC), total flavonoid content (TFC) and antioxidant activities of each fraction from GBE.

GBE	TPC (mg GAE/g Dry Weight)	TFC (mg CAE/g Dry Weight)	DPPH Scavenging (SC_50_)	Iron Chelation (EC_50_)
CME	358.88 ± 17.09	232.58 ± 6.83	-	-
HSF	13.48 ± 2.26	24.57 ± 1.77	-	-
ESF	298.72 ± 7.82	172.61 ± 5.16	35.73 ± 2.17	146.00 ± 16.52
EIF	303.02 ± 1.87	201.25 ± 4.93	41.24 ± 5.81	106.67 ± 2.89

**Table 3 foods-12-02648-t003:** The chemical contents of each fraction from GBE.

GBE	Content (mg/100 g Extract)						
	Chlorogenic Acid (CGA)*	Gallic Acid (GA)	Catechin Gallate (CAT)	Epicatechin (EPI)	Caffeine (CF)*	Gallocatechin Gallate (GCG)	Epicatechin Gallate (ECG)
CME	17,768.36 ± 14.29	UD	132.60 ± 0.38	UD	7033.88 ± 60.34	93.25 ± 1.61	UD
HSF	41.10 ± 0.32	2.15 ± 0.21	UD	UD	240.69 ± 0.91	UD	UD
ESF	2845.18 ± 9.31	98.02 ± 4.62	UD	UD	61,043.30 ± 486.96	UD	UD
EIF	17,683.78 ± 19.51	UD	133.35 ± 0.64	UD	6589.86 ± 10.18	105.99 ± 2.69	UD

UD = Undetectable.

## Data Availability

Data is contained within the article or Appendix A.

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
