# Peer review of "Inhibitory Effects of Chlorogenic Acid Containing Green Coffee Bean Extract on Lipopolysaccharide-Induced Inflammatory Responses and Progression of Colon Cancer Cell Line"

_foods, 2023, doi:10.3390/foods12142648_

Round 1
Reviewer 1 Report
Foods Review June 2023 – Green Coffee Bean
This paper reports on the potential of various extracts of green coffee beans for aiding in the treatment of colorectal cancer. The work is interesting and carefully carried out and is well written.
Comments:
1. L 77-78 – was the filtrate evaporated to dryness or to a known volume? I assume it is to a known volume (not stated) since it was extracted with hexane.
2. The authors could explain the rationale for the sequential extraction procedure followed. What is the reason for not extracting the coffee with hexane and ethyl acetate separately, keeping in mind that there will be little hydrophobic compounds in the methanol extract.
3. L 79 – “ration” should be “ratio”?
4. L 154 – what is “wo”?
English language is satisfactory.
Author Response
We would like to express our sincere gratitude for dedicating your valuable time to thoroughly review our manuscript. Your insightful suggestions and feedback have immensely contributed to the refinement of this work. We have diligently incorporated your recommendations, resulting in a significant enhancement of the overall quality of this article
Responses to Reviewer 1
This paper reports on the potential of various extracts of green coffee beans for aiding in the treatment of colorectal cancer. The work is interesting and carefully carried out and is well written.
- L 77-78 – was the filtrate evaporated to dryness or to a known volume? I assume it is to a known volume (not stated) since it was extracted with hexane.
Author response: Thank you for your attention to detail on this point. We already edited the extraction procedure and informed the final volume of CME (after evaporation)
“The filtrate was collected by Whatman No.1 and then concentrated to a final volume of 100 mL by the evaporator, called crude methanol extract (CME).” (line 75-77)
- The authors could explain the rationale for the sequential extraction procedure followed. What is the reason for not extracting the coffee with hexane and ethyl acetate separately, keeping in mind that there will be little hydrophobic compounds in the methanol extract.
Author response: We agree that the rationale for the sequential extraction procedure needs to be demonstrated. We already edited the extraction procedure and informed the rationale of hexane and ethyl acetate extraction
“Thereafter, CME (100 mL) was further separated with hexane to remove lipophilic fraction (in ratio 1:1) into 2 parts; hexane soluble fraction (HSF) and hexane insoluble fraction (HIF). HIF was further separated with ethyl acetate to decaffeinate (in ratio 1:1) into 2 parts; ethyl acetate soluble fraction (ESF) and ethyl acetate insoluble fraction (EIF).” (line 77-81)
- L 79 – “ration” should be “ratio”?
Author response: This point was already corrected (line 78)
- L 154 – what is “wo”?
Author response: wo is stand for without. However, we already changed wo to without for clearer sentence. (line 152)
Thank you once again for your meticulous assessment and valuable input.
Reviewer 2 Report
Title:
Chlorogenic acid-enriched fraction from green coffee bean ex- 2 tract suppresses inflammatory responses and cancer progres- 3 sion on lipopolysaccharide-induced human colorectal cancer, 4 through the TLR4-NF-κB pathway
Plagiarism : Reduce the plagiarism up to 15%
Abstract: Minimized the Abstract
Introduction: Add some more part related to the green coffee bean and human colorectal cancer and human gut microbiota with latest references.
recommended references.
1. Pabari, K.; Pithva, S.; Kothari, C.; Purama, R.K.; Kondepudi, K.K.; Vyas, B.R.M.; Kothari R.; Ambalam, P. Evaluation of probiotic properties and prebiotic utilization potential of Weissella paramesenteroides isolated from fruits. Probiotics and Antimicrobial Proteins 2020, 12(3), 1126–1138. doi:10.1007/s12602–019–09630–w
2. Abid, Rameesha, Hassan Waseem, Jafar Ali, Shakira Ghazanfar, Ghulam Muhammad Ali, Abdelbaset Mohamed Elasbali, and Salem Hussain Alharethi. "Probiotic Yeast Saccharomyces: Back to Nature to Improve Human Health." Journal of Fungi 8, no. 5 (2022): 444.
Methodology
1. Improve the technical writing of methods by using the latest methods
2. green coffee bean extract methods must be recheck
Discussion:
Improve the discussion part, add latest references.
References should be journal pattern
Conclusions must be shorter
Nill
Author Response
We would like to express our sincere gratitude for dedicating your valuable time to thoroughly review our manuscript. Your insightful suggestions and feedback have immensely contributed to the refinement of this work. We have diligently incorporated your recommendations, resulting in a significant enhancement of the overall quality of this article
Responses to Reviewer 2
Plagiarism : Reduce the plagiarism up to 15%
Author response: We already checked and reduced plagiarism (proved by Grammarly online program)
Abstract: Minimized the Abstract
Author response: We already revised abstract from 200 words to 159 words (line 14-25)
Introduction: Add some more part related to the green coffee bean and human colorectal cancer and human gut microbiota with latest references.
recommended references.
- Pabari, K.; Pithva, S.; Kothari, C.; Purama, R.K.; Kondepudi, K.K.; Vyas, B.R.M.; Kothari R.; Ambalam, P. Evaluation of probiotic properties and prebiotic utilization potential of Weissella paramesenteroides isolated from fruits. Probiotics and Antimicrobial Proteins 2020, 12(3), 1126–1138.doi:10.1007/s12602–019–09630–w
- Abid, Rameesha, Hassan Waseem, Jafar Ali, Shakira Ghazanfar, Ghulam Muhammad Ali, Abdelbaset Mohamed Elasbali, and Salem Hussain Alharethi. "Probiotic Yeast Saccharomyces: Back to Nature to Improve Human Health." Journal of Fungi 8, no. 5 (2022): 444.
Author response: New references about green coffee bean and its major compounds related to gram negative pathogenic bacteria (especially in colon) were added (line 65-66) and correctly cited in the text.
- Canci, L.A.; Benassi, M.T.; Canan, C.; Kalschne, D.L.; Colla, E. Antimicrobial potential of aqueous coffee extracts against pathogens and Lactobacillus species: A food matrix application. Food Biosci. 2022, 47, doi:10.1016/j.fbio.2022.101756.
- Bharath, N.; Sowmya, N.K.; Mehta, D.S. Determination of antibacterial activity of green coffee bean extract on periodontogenic bacteria like Porphyromonas gingivalis, Prevotella intermedia, Fusobacterium nucleatum and Aggregatibacter actinomycetemcomitans: An in vitro study. Contemp. Clin. Dent. 2015, 6, 166-169, doi:10.4103/0976-237X.156036.
- Sales, A.L.; dePaula, J.; Mellinger Silva, C.; Cruz, A.; Lemos Miguel, M.A.; Farah, A. Effects of regular and decaffeinated roasted coffee (Coffea arabica and Coffea canephora) extracts and bioactive compounds on in vitro probiotic bacterial growth. Food Funct. 2020, 11, 1410-1424, doi:10.1039/c9fo02589h.
Methodology
- Improve the technical writing of methods by using the latest methods
Author response: We already edited the part of materials and methods.
- green coffee bean extract methods must be re check
Author response: We agree that the rationale for the sequential extraction procedure needs to be demonstrated. We already edited the extraction procedure and informed the rationale of this extraction procedure. The extraction procedure has been revised. (line 72-83)
Discussion:
Improve the discussion part, add latest references.
Author response: The discussion part has been revised according to your suggestion and references were updated (mostly within 10 years ago). Even though some references were old, they were quite necessary to refer. (line 392-463)
References should be journal pattern
Author response: The references were re-organized according to MDPI pattern (line 485-588)
Conclusions must be shorter
Author response: We already revised and removed some part of conclusions from104 words to 88 words (line 464-470)
Thank you once again for your meticulous assessment and valuable input.
Reviewer 3 Report
the work is interesting and complete
Author Response
We would like to express our sincere gratitude for dedicating your valuable time to thoroughly review our manuscript. We are very glad that this manuscript has been satisfactorily written in your opinion. However, we have diligently incorporated the recommendations from al reviewer and editor, resulting in a significant enhancement of the overall quality of this article.
Reviewer 4 Report
This is a comprehensive work describing the effects of a fraction og green coffee beans on cancer treatment and increasing the efficacy of drugs. All possible aspects have been covered, manuscript id well-prepared and presented.
Author Response

(The authors gave the same response as above.)
